# Event-Related Potential to Conscious and Nonconscious Emotional Face Perception in Females with Autistic-Like Traits

**DOI:** 10.3390/jcm9072306

**Published:** 2020-07-21

**Authors:** Vilfredo De Pascalis, Giuliana Cirillo, Arianna Vecchio, Joseph Ciorciari

**Affiliations:** 1Department of Psychology, La Sapienza University of Rome, 00185 Rome, Italy; giuliana.cirillo.91@gmail.com (G.C.); arianna.vecchio@uniroma1.it (A.V.); 2Centre for Mental Health, Department of Psychological Sciences, Swinburne University of Technology, Hawthorn, VIC 3122, Australia; jciorciari@swin.edu.au

**Keywords:** Autism Spectrum Quotient, ERPs, face perception, emotion

## Abstract

This study explored the electrocortical correlates of conscious and nonconscious perceptions of emotionally laden faces in neurotypical adult women with varying levels of autistic-like traits (Autism Spectrum Quotient—AQ). Event-related potentials (ERPs) were recorded during the viewing of backward-masked images for happy, neutral, and sad faces presented either below (16 ms—subliminal) or above the level of visual conscious awareness (167 ms—supraliminal). Sad compared to happy faces elicited larger frontal-central N1, N2, and occipital P3 waves. We observed larger N1 amplitudes to sad faces than to happy and neutral faces in High-AQ (but not Low-AQ) scorers. Additionally, High-AQ scorers had a relatively larger P3 at the occipital region to sad faces. Regardless of the AQ score, subliminal perceived emotional faces elicited shorter N1, N2, and P3 latencies than supraliminal faces. Happy and sad faces had shorter N170 latency in the supraliminal than subliminal condition. High-AQ participants had a longer N1 latency over the occipital region than Low-AQ ones. In Low-AQ individuals (but not in High-AQ ones), emotional recognition with female faces produced a longer N170 latency than with male faces. N4 latency was shorter to female faces than male faces. These findings are discussed in view of their clinical implications and extension to autism.

## 1. Introduction

Autism is a neurodevelopment condition involving dysfunction in reciprocal-social interaction. Deficits in decoding and understanding facially expressed emotions occur commonly in autism spectrum disorders ASDs; see [1], which contribute to the impairment of social communication that serves as one of its core diagnostic criteria; for a review see [2]. Several difficulties in the processing of facial expressions have been reported with ASD [3,4,5,6,7,8] and their relatives [9,10,11,12,13,14]. Subclinical traits of autism are observed in the general population (i.e., meeting a diagnosis of autism) and are represented by extreme values on a continuous distribution [15]. Autistic-like traits constitute potential markers of family genetic liability to autism [16,17,18]. The Autism Spectrum Quotient (AQ) [19] has been developed to measure the degree to which an adult with normal intelligence has autistic traits with a threshold score of 26 to meet a diagnosis of autism [20]. Emotional processing deficits in orienting [21], visual facial scanning [22,23], and in cognitive evaluation of facial expressions are also reported in autism. If individuals with ASD exhibit dysfunctional neural activity in response to emotional faces [5,24,25], perhaps individuals with high AQ (Hi-AQ) may as well.

Neuroimaging research has outlined the central role of amygdala in the processing of facial emotions in non-clinical populations, including fearful and non-threatening facial expressions [26,27,28,29]. However, few studies have evaluated the neural correlates of sadness recognition. Blair and colleagues [30] showed that viewing sad facial expressions activated the left amygdala and right temporal pole. Viewing sad films resulted in activations in a network including the medial prefrontal cortex, superior temporal gyrus, precuneus, lingual gyrus, and the amygdale [31]. A deficit in sadness recognition could therefore be explained by disrupted amygdala-cortical connectivity [8]. The amygdala plays an essential role in a vigilance system for rapidly alerting other brain regions to the importance of social stimuli. When emotional faces are presented under 40 ms and immediately followed by a neutral “backwards masking” face, participants reported no awareness of the emotional face but demonstrated increased right amygdala activation [32]. The backward masking paradigm examines subliminal automatic processes along the subcortical route [33,34] and is highlighted by event-related potentials (ERPs; e.g., [35,36,37]. Behavioral study findings suggest that individuals with ASD are less affected by nonconscious information compared to typically developing (TD) controls [38,39]. 

Fujita and coworkers [40] measuring visual evoked potentials (VEPs) found that the N1 component of VEPs elicited by chromatic gratings (that preferentially activate the parvocellular (P) color pathway) was significantly prolonged in ASD participants compared to TD controls, suggesting that ASD involves dysfunction of the P-color pathway at a relatively low level of information processing. In a later study [41], VEPs in TD and high-functioning ASD subjects were elicited by using the backward-masking paradigm with subliminally presented fearful and neutral faces and objects (upright and inverted positions). In the TD group, the N1 amplitude for the subliminal upright fearful (but not neutral) faces was found significantly higher than the inverted ones, while in the high-functioning ASD subjects this measure did not show this effect. These findings indicated altered early visual processing of short duration emotional faces in ASD.

The dimensional approach to understanding personality disorders offers the possibility of studying particular trait aspects of the ASD syndrome in TD individuals who may not satisfy diagnostic criteria for the disorder but may embody particular components of the syndrome [42]. ERP research has highlighted the N170 wave as a face-specific component reflecting the earliest stages of face processing [43,44]. However, there is no reliable experimental support for the modulation of the N170 wave by emotional facial expressions in TD subjects see e.g., [45,46,47], as well as in subjects with autism [48,49,50,51]. Yet, the N170 wave does differentiate children and adults with autism compared to those without autism [49,52,53,54]. In addition to the N170 wave, other studies have demonstrated that N1 wave has been found significantly modulated by affect in the early phase of facial perception processing, with greater negativity for fearful versus neutral faces [55,56]. Research has also reported an increased N2 for fearful compared to non-fearful faces at supraliminal and supraliminal stimulation [36,57], and subliminal orienting and automatic aspects of face processing [58]. The N2 together with late P3 wave can discriminate, respectively, subliminal and supraliminal fearful face-processing. Subliminal condition has been distinguished by the enhanced N2 wave (“excitatory”) to fearful faces, representing the orienting and automatic aspects of face processing. Supraliminal perception of fearful faces was distinguished by the enhanced late P3/N4 wave (“inhibitory”), representing the integration of emotional processes [58,59]. 

The N4 has been linked to conscious perception of emotional faces. It was found to enhance together with the late P3 in response to supraliminal fear perception at the parietal midline site and thought to be involved specifically with the controlled integration and cognitive elaboration of facial emotional information, [58,60]. More details on the empirical and theoretical aspects of the P300 are reported by Polich [61]. In his overview of the P300 theory, he dissected the P300 into its constituent frontal P3a (early-P3) and temporal-parietal P3b (late-P3) and outlined how the P3a and P3b may interact. He inferred that stimulus evaluation engages first focal attention (P3a) to facilitate context maintenance (P3b), which is associated with memory storage operations that are then initiated in the hippocampal formation with the updated output transmitted to parietal cortex; a late P3b is produced to establish the connection with storage areas in associational cortex.

More recently, Stavropoulos et al. [62] conducted an ERP study to evaluate the relationship of AQ trait with face efficiency processing under subliminal and supraliminal conditions in TD adults. Regardless of AQ score, these authors obtained higher P1 and P3 amplitudes and shorter N170 latencies for nonconscious versus consciously presented faces. In addition, High AQ (Hi-AQ) traits were associated with delayed ERP components, indicating that inefficient face perception is present in individuals with subclinical levels of social impairment. In this research line, Vukusic and coworkers [63] elicited ERPs by a backward-masked paradigm under subliminal and supraliminal conditions to evaluate the sensitivity of ERP waves to differentiate conscious and nonconscious information processing for neutral, fearful and happy faces in Hi- AQ and low AQ (Lo-AQ) traits. These authors found partial support for their main hypothesis that differences between AQ groups would emerge in emotion effect under subliminal viewing conditions, since they found enhanced frontal N2 amplitude only for subliminally presented happy faces in the Lo-AQ, but not in the Hi-AQ group. They obtained shorter late ERP components of frontal P3 and N4 latencies (representing event integration) for subliminal vs. supraliminal condition. Finally, they also disclosed shorter N170 latencies for supraliminal vs. subliminal conditions across both AQ groups, although they did not observe any group differences on the face-specific N170 component. 

According to the literature, typically, mothers spend more time in direct face to face contact with their young children than fathers do. This in turn may affect the child’s experience of faces, thus facilitating the development of skills for accurate face processing [64] and to discriminate mother from stranger [65,66]. Infants later diagnosed with autism may fail strongly to attend to faces discrimination from those who were TD infants [7]. Thus, we hypothesize that female faces should capture more attention than male faces mainly in Lo-AQ participants, while these differences should be less pronounced in Hi-AQ participants. 

To our knowledge, there have been few reports showing specific deficits for sadness recognition in the autistic population, let alone in an all-female cohort. Exceptions are the behavioral study by Boraston et al. [67], supporting evidence for impaired sad face recognition in adults with autism, and the ERP study by O’Connor et al. [53] reporting delayed N170 latencies to sad faces and face parts in adults with ASD. Stavropoulos and colleagues [62] reported larger P3 amplitudes and shorter N170 latencies for nonconscious versus consciously presented faces and delayed ERP components in Hi-AQ scorers for fear and neutral faces, but they did not find significant differences in their N170 latency to sad faces.

### 1.1. Aims

In the current study, we used a subliminal and supraliminal presentation of faces through a backward-masking paradigm similar to that used in the Vukusic’s and colleagues’ experiment [63], but with the new inclusion of the gender face factor in the recognition of emotional facial expressions. This was done to extend ERP results of the referenced study to facial-gender (female, male) factor and the interaction of this factor with the emotional expression among different levels of AQ traits. As far as we know, in the current literature, the influence of facial gender on the recognition of emotional facial expressions has been overlooked. Thus, in addition to the widely used happy and neutral faces, a further aim of the present study was to further test differences in Hi- and Lo-AQ trait to sad faces. Finally, considering that ASD is a predominantly male disorder known to manifest sex differences in face perception [68], our current investigation was limited to the analyses of AQ score in a TD female cohort.

### 1.2. Hypotheses

In terms of behavioral performance, our primary hypotheses were: (1a) Hi-AQ participants should be less accurate than Lo-AQ ones to recognize emotions of happy, neutral, and sad facial expressions [69], and (1b) this difference should be more pronounced for subliminal stimuli [38,39]. (1c) Lo-AQ scorers should have a higher accuracy in the recognition of facial expressions with female than male faces, while these facial-gender differences should be less pronounced in Hi-AQ scorers [7]. 

In terms of ERP waves, our main hypotheses were: (2) to find prolonged N1 latency within the occipital cortex in Hi-AQ relative to Lo-AQ participants; this hypothesis was done to extend to the general population previously N1 latency findings in ASD individuals [40]. (3) Hi-AQ, relative to Lo-AQ participants, would exhibit a smaller N170 amplitude and/or a delayed N170 latency [54,62], and (4) sad faces should evoke a larger N170 peak amplitude [70] with longer N170 latency than happy faces [71,72]. (5) In line with Vukusic and colleagues findings, Lo-AQ relative to Hi-AQ scorers should show enhanced N2 amplitude for subliminal happy faces, and Low-AQ showing a shorter frontal late P3 and N4 latencies for subliminal vs. supraliminal condition [63].

This study was carried out to investigate whether previous findings of reduced sensitivity to subliminal and supraliminal emotional faces in ASD are limited to individuals with ASD, or whether these findings can be extended in a sample of TD female adults characterized by higher scores of autistic-like traits.

## 2. Methods

### 2.1. Participants

Fifty right-handed neurotypical female students volunteered (informed consent) to participate (18–31 years; mean age = 23.0, SD = 3.0). The study was approved by the institutional review board of the Department of Psychology at La Sapienza University of Rome in accordance with the Helsinki Declaration. All participants had normal or corrected-to-normal visual acuity, were medication-free, and had no reported history of either psychiatric or neurological disorders (including clinical autism).

### 2.2. Personality Measures

The sample consisted of 50 neurotypical right-handed women students recruited through local advertisements. All participants completed a battery of personality questionnaires in a session preceding the electrophysiological recordings. Hand preference was assessed with the Italian version of the Edinburgh Handedness Inventory [73]. The personality measures of interest in this study were:
(1)Autism Spectrum Quotient (AQ). The AQ is a self-administered questionnaire consisting of 50 questions, devised to quantitatively measure the degree to which a person with normal intelligence has autistic traits [19,20]. Participants respond using a 4-point rating scale (definitely agree—slightly agree—slightly disagree—definitely disagree) across five domains: social skills, attention switching, attention to detail, communication, and imagination. The individual scores one point for each answer that reflects abnormal or autistic-like behavior. This measure is sensitive to autistic traits in nonclinical populations [74,75]. (2)The Raven’s Advanced Progressive Matrices (RAPM). The RAPM is a standardized nonverbal intelligence test and is generally used as a test of general cognitive ability and intelligence [76]. It consists of visually presented geometric figures where one part is missing, and the missing part must be selected from a panel of suggested answers to complete the designs. The RAPM was used to eliminate general intelligence as a potential explanation of any differences found between AQ groups. On this basis, all participants had a RAPM score of at least 14, which is in the normal range of the Italian Population (*M* = 20.4, *SD* = 5.6, Age range: 15–47 years, N = 1762) [76].

### 2.3. Stimuli

The identical face stimuli protocol reported in the Vukusic et al., study [63] and Goodin et al., study [77] was used. The stimuli consisted of colored photographs of the faces of four Caucasian models (equal male & female) depicting neutral, happy and sad expressions (with closed mouth exemplars) and were cropped with an oval shape. Faces were selected from the NimStim collection (http://www.macbrain.org/resources.htm), a freely available collection of emotional face stimuli with good internal validity and reliability [46]. Best attempts were made to carefully match the stimuli sets (faces) on a variety of variables that may affect attentional processes including luminance, color and contrast. This was done with Photoshop CS2, which makes it possible to equate luminance and contrast across the different emotional expressions. The fills (masks) were made using Adobe Photoshop CS2 (http://www.photoshop.com) and contained two colors, purple or yellow, which were selected due to their color opposition. In an attempt to infuse a different visual identity into the fills similar to the differing identities seen in the faces, the two gratings and one pattern consisted of varying widths ranging from 5 mm line widths to 12 mm in 1 mm increments. Faces were contained within a black border to focus the participant’s attention on the characteristics of the faces presented and not peripheral characteristics such as hair or ears. Fills were also presented within a black border. This was done to reduce the risk of low-level changes in these properties influencing the early ERPs. The techniques employed to control for these changes were based on the Willenbockel, Sadr [78] method. All pictures were color photographs (visual angle: 7.4° × 5.1°; mean luminance: 22.5 cd/m^2^, with a viewing distance of 100 cm). We used an oval purple/black chromatic square-wave grating as a pattern mask of the same luminance of photographs (spatial frequency of 0.6 cycles per degree), which was surrounded by a homogeneous black color background.

To identify the sub-threshold duration at which participants would be able to determine whether the masked stimuli were faces, we invited 10 psychology students (20 to 34 years, M = 23.6, SD = 2.4 years) in a pilot experiment. We used an ascending series of trials to prevent participants from perceiving the contents of the stimuli. In each trial, masked stimuli (neutral, happy, sad faces and mask images) were randomly presented, and participants verbally reported what they saw. In the first trial block, the stimulus presentation was 10 ms long and increased by 10 ms steps in each subsequent trial block. Stimuli were presented 20–30 times in each trial block. The threshold at which participants first reported that they saw a face-like shape ranged between 20 and 60 ms, with a mean of 45.8 ms. Based on the results of this experiment, we set the duration of sub-threshold presentation in the current study at 21 ms. The study was conducted in line with previous ERP studies cited in the text. Moreover the blank screen was set at 847 ms, due to trying to move away from EEG effects associated with kindling and, therefore, attempt to induce a “cleaner” ERP [63]. 

### 2.4. Procedure

Participants sat in a dimly lit, sound and electrically shielded booth in front of a computer screen. Stimuli were presented on a 19” color LCD monitor (1400 × 900 resolution and 75 Hz vertical refresh rate) and in 8 blocks of 120 trials; each block consisting of a randomized presentation of both subliminal and supraliminal emotional (positive, negative, and neutral) female and male faces. Block order was counterbalanced across participants with an equal number of trials in each condition for each facial expression (120 trials for each facial expression, for each condition).

The faces task (in E-Prime 2.0) began with a central white fixation cross followed by a picture of a face stimulus, which was displayed for duration of 21 ms (subliminal) or 167 ms (supraliminal). At the end of each trial, a question appeared on the screen asking for explicit emotion recognition for each face; numbers 1–3 (1 = neutral, 2 = happy, 3 = sad; Figure 1), allowing unlimited time to respond with their right hand. In the case of subliminal stimuli, participants were asked to guess the facial expression. The explicit recognition task was adopted because it gives equal importance to all facial expressions in both conditions. 

### 2.5. EEG Recording

EEG and Electro-ocular (EOG) were acquired using a 40-channel NuAmps DC amplifier system (Neuroscan Acquire 4.3, Compumedics Neuroscan Inc, Char lotte, North Carolina 28269, USA). Signals were band-limited to 75 Hz (and 50 Hz notch filter), the gain was set at 200, and the sampling rate was 1000 Hz, with impedances under 5 kΩ. Standard tin electrodes with electrolyte gel were used. Bipolar horizontal and vertical EOG were recorded, respectively, from the epicanthus of the right and the left eye and from the supra- and infra-orbital positions of the left eye. EEG was recorded from 30 electrodes (i.e., Fp1, Fp2, F7, F8, F3, F4, FT7, FT8, T3, T4, FC3, FC4, C3, C4, CP3, CP4, TP7, TP8, T5, T6, P3, P4, O1, O2, Fz, FCz, Cz, CPz, Pz, Oz) by an electro-cap using an extended montage of the standard 10–20 system. The referenced electrode was obtained by linked ears (A1 + A2)/2 with a ground electrode placed 10 mm anterior to Fz. EEG data were analyzed offline using the Brain Vision Analyzer 2.1.0 (Brain Products GmbH, Gilching, Germany). E-Prime 2.0 (Psychology Software Tools, Inc., Sharpsburg, PA, USA) served to deliver auditory stimuli and triggers for EEG recordings. The resting EEG was recorded during eyes open and closed for 3 min.

EEG data was analyzed offline using the Brain Vision Analyser for preprocessing and eye movement correction procedures. Each recording epoch (1000 ms) included a baseline of 100 ms before stimulus onset. Eye blink correction was first performed [79] and residual artifact exceeding ±75 µV was removed. Recordings were re-referenced to the average reference as computed from all scalp electrodes (for the N170 component), while an earlobes’ reference was used for N1, P2, N2, and P3 components, for endogenous ERP components. The common average reference was used as it yielded the largest N170 amplitude [63,80]. ERPs were averaged separately for each stimulus category (each emotion was averaged for subliminal and supraliminal threshold conditions) and baseline corrected. 

### 2.6. Behavioral Data Analysis

We measured the accuracy rates (percent of correct detections of the facial expressions) and due to unlimited time given to participants for recognizing facial expressions, we did not report the reaction times. To test the effect of gender and AQ factor on task performance, the accuracy scores were compared using an ANCOVA with Emotion, Condition and Facial-Gender as within-subject factors, while AQ scores were used as a covariate. Two-tailed *t*-tests were used to compare accuracy to chance levels for the two conditions and for each of the emotions within the conditions. To control for false-positive errors, significance levels for *F* and *t*-test coefficients were corrected by using the false discovery rate (FDR) method [81].

### 2.7. ERP Analyses

ERP components (target) were identified and quantified across Fz, Cz, Pz, and Oz midline sites. The following components were identified: N1 ERP peak was (M ± SD = 104.0 ± 5.0 ms), quantified as negative values as the baseline-to-peak difference in voltage for the most negative peak within the 90–140 ms window following face stimulus onset; N2 (217.7 ± 9.5 ms, window 170–310 ms); P3 (320 ± 12.4 ms, window 200–390 ms); N4 (382.2 ± 9.2 ms, window 330–500 ms). Finally, the N170 ERP wave was also examined at the lateral posterior-temporal sites T5 and T6 (closest to the occipito-temporal sites P7 and P8). This ERP component was peaking at 180.7 ± 13.1 ms and was measured within the 140–260 ms time window [35,36,58,63]. Peak values were first semi-automatically detected as local minima for negative waves (or maxima, for positive waves) and then, after visual inspection, the position of the peak changed manually if necessary. The N1, N2, and N170 peak values were then multiplied by -1 and expressed as positive values for our convenience. The ERP analysis included both correct and incorrect behavioral responses. The amplitude and latency of each ERP component were quantified by the highest peak value within the chosen latency window. ERP amplitude and latency were analyzed with repeated measures ANCOVAs using AQ scores as a covariate. For these analyses, Emotion (neutral, happy, sad), Condition (subliminal, supraliminal), Facial-Gender (female, male), and Electrode Location (Fz, Cz, Pz, Oz) as within-subject factors were used. For the N170 wave, Electrode Location was replaced with Hemisphere (T5, T6) factor. An alpha criterion level of 0.05 was used unless otherwise noted. Huynh-Feldt adjustments were used when the assumption of sphericity was violated [82]. 

To report effect size estimates, partial ɳ^2^_p_ values (see Appendix A for statistical details) were also calculated. Paired samples *t*-tests were performed to supplement the behavioral and ERP findings. To control for false-positive errors, significance levels obtained for ERP measures were corrected by applying the false discovery rate correction (FDR) method across all ERP amplitude and latency measures [81]. Only for graphical illustrations, and to understand the direction of changes of significant main and/or interaction effects involving AQ trait, we applied a separate median split on this personality measure (M = 14.9, Md = 14.5; Skewness = 0.278, Kurtosis = −0.984). Participants were considered as belonging to either group Hi-AQ (N = 25, M = 20.6, SD = 4.3, Range = 15–26) or Lo-AQ (N = 25, M = 9.2, SD = 2.8, Range = 3–14) when their AQ scores were above or below the median. None of the AQ scores fell on the median, therefore, none of the participants were excluded.

## 3. Results

### 3.1. Behavioral and Personality Results

Pearson correlation coefficients among trait measures of interest together with descriptive statistics are reported in Table 1. There was no evidence for a significant relation of AQ with RAPM (*p* > 0.05). There were also no significant between AQ group differences on the RAPM (Hi-AQ: M = 21.8, SD = 5.2; Lo-AQ: M = 22.7, SD = 5.1; t = −0.66, *p* = 0.51). 

The repeated measure ANCOVA on accuracy scores yielded a significant main effect for AQ (F (1,48) = 25.82, *p* < 0.001, ɳ^2^_p_ = 0.349), showing a lower accuracy in Hi-AQ scores compared to Lo-AQ ones (i.e., M = 71.0%, SD = 6.5 vs. M = 76.5%, SD = 0.04). The main effect of Condition (F (1,48) = 277.62, *p* < 0.001, ɳ^2^_p_ = 0.853) and the AQ × Condition interaction (F (1,48) = 7.93, *p* = 0.0137, ɳ^2^_p_ = 0.144) were both significant. The first effect indicated lower accuracy rates in the subliminal compared to supraliminal condition (M = 55.3, SD = 7.0 vs. M = 92.3%, SD = 7.5); the second effect showed that recognition accuracy of supraliminal faces in Hi-AQ participants was significantly lower than that in Lo-AQ ones (M = 88.2%, SD = 8.5% vs. M = 96.3%, SD = 3.1, *p* < 0.001), while between-group difference of subliminal faces did not reached the significance level (M = 53.8%, SD = 6.5% vs. M = 56.8%, SD = 7.3, *p* = 0.06). Further, Facial-Gender factor (F (1,48) = 23.83, *p* < 0.001, ɳ^2^_p_ = 0.331) and the Facial-Gender × AQ interaction (F (1,48) = 61.18, *p* < 0.001, ɳ^2^_p_ = 0.560) were both significant. The first effect disclosed a higher accuracy rate for female than male faces (M = 76.3%, SD = 4.5% vs. M = 71.2%, SD = 10.1%). The interaction effect indicated that Hi-AQ participants’ accuracy to recognize facial expression with female faces was significantly higher than that with male faces (M = 76.5%, SD = 5.3%, vs. M = 65.6%, SD = 10.5%, *p* < 0.001), while in Lo-AQ there were no differences in facial expression recognition with female versus male faces (M = 76.2%, SD = 3.5%, vs. M = 76.9%, SD = 5.5%, *p* > 0.05). 

This effect also indicated that Lo-AQ individuals, as compared with the Hi-AQ ones, had a significantly higher accuracy for male faces (*p* < 0.001), but not for female faces (*p* > 0.05). Finally the fourth order interaction of Facial-Gender × Condition × Emotion × AQ was significant (F (2,96) = 2.99, *p* < 0.05, ɳ^2^_p_ = 0.057) and disclosed that Hi-AQ participants had higher accuracy in detecting emotions with female faces than male faces for both subliminal and supraliminal stimuli. In contrast, in Lo-AQ participants, facial gender differences did not reach the significance level (Figure 2). On the whole, these findings support our first hypothesis (more statistical details are available as Appendix A).

### 3.2. ERP Results

#### 3.2.1. N1 Amplitude and Latency

The ANCOVA, using AQ scores as a covariate, on the N1 amplitude data showed a significant main effect of Location (F (3,144) = 13.64, *p* < 0.001, ɳ^2^_p_ = 0.221), indicating that the N1 was larger at frontal (Fz) than parietal (Pz) and occipital (Oz) regions (both *p* < 0.0001) and also larger at central (Cz) than Pz and Oz (both *p* < 0.001) regions. Further, the significant Emotion × Location interaction (F (2,288) = 5.11, *p* = 0.0022, ɳ^2^_p_ = 0.096) demonstrated larger N1 amplitudes for happy than neutral and sad faces at Fz, Cz and Pz sites (Fz: M = 7.2, SD = 3.0; M = 3.6, SD = 1.5; M = 3.3, SD = 1.5; Cz: M = 6.6, SD = 3.2; M = 3.4, SD = 1.5; M = 3.4, SD = 1.5; Pz: M = 2.5, SD = 2.4; M = 1.3, SD = 1.3; M = 1.3, SD = 1.1, respectively for happy, neutral, and sad faces; all ts, *p* < 0.001). In addition, the significant Emotion × Location × AQ interaction (F (6,288) = 4.48, *p* = 0.0046, ɳ^2^_p_ = 0.083). Simple effect analysis at each recording site indicated that in the Lo-AQ group there was a larger negative peak for sad compared to happy and neutral expressions at Pz, and P4 leads (Pz: sad vs. happy t = 2.30, *p* < 0.05; sad vs. neutral t = −2.21, *p* < 0.05; P4: sad vs. happy t = 3.55, *p* < 0.01; sad vs. neutral t = −2.17, *p* < 0.05; paired *t*-tests respectively for sad vs. happy and sad vs. neutral faces). In contrast, paired samples *t*-tests performed separately in the Hi-AQ groups did not disclose any significant difference between emotions (all *p* > 0.05; see right quadrant of Figure 3).

The ANCOVA on N1 latency disclosed a main effect of Location (F (3,144) = 49.80, *p* = 0.0019, ɳ^2^_p_ = 0.509) and of Condition (F (1,48) = 6.65, *p* = 0.015, ɳ^2^_p_ = 0.122). The first effect showed a progressive significant reduction in N1 latency starting from Fz to Cz, Pz, and Oz sites (all *p* < 0.001; see Table 2). The second main effect indicated that subliminal stimuli elicited shorter N1 latencies than supraliminal stimuli (Table 2). Moreover, the significant AQ × Emotion interaction (F (2,96) = 4.86, *p* = 0.0123, ɳ^2^_p_ = 0.092) showed that in Hi-AQ participants happy faces had a longer N1 latency than neutral and sad faces (M = 106.2, SD = 4.5 vs. M = 104.2, SD = 4.6, *p* < 0.05 and vs. M = 104.5, SD = 4.9, *p* < 0.05), while, in contrast, in Lo-AQ there were no differences among emotional faces (M = 102.3, SD = 6.2 vs. M = 103.5, SD = 6.2, and M = 103.3, SD = 6.2, all *p* > 0.05). In addition, Hi-AQ had a longer N1 latency to happy faces compared to Lo-AQ participants (M = 106.2, SD = 4.5 vs. M = 102.3, SD = 6.2, *p* < 0.05), whilst there were no latency differences between AQ groups for the neutral and sad faces. A simple analysis conducted on N1 amplitude data of the Oz lead alone found a main effect of AQ (F (1,48) = 4.95, *p* = 0.031, ɳ^2^_p_ = 0.093), indicating a relatively longer N1 latency at occipital midline region in the Hi-AQ scorers. This finding was in support of our second main hypothesis and in line with Fujita, Yamasaki [40] findings in ASD patients (see Figure 4).

Finally, the interaction effect of Facial-Gender × Emotion × Location (F (6,288) = 4.91, *p* < 0.001, ɳ^2^_p_ = 0.093) and Facial-Gender × Emotion × Location × AQ (F (6,288) = 3.10, *p* = 0.0123, ɳ^2^_p_ = 0.061) were both significant. These effects disclosed that for female happy faces, Hi-AQ scorers had a longer N1 latency than Lo-AQ scorers at Fz and Oz scalp leads, while for female sad faces, this between-group difference was significant for the Fz lead alone. For male happy and sad faces, there was also a relative longer N1 latency in Hi-AQ scorers, although this difference was significant at only the occipital lead (see Figure 4).

#### 3.2.2. N170 Amplitude and Latency

The analysis on the N170 amplitude data showed a significant effect of Condition (F (1,48) = 7.99, *p* = 0.0123, ɳ^2^_p_ = 0.142), which was due to a larger N170 peak to supraliminal than subliminal faces (Figure 5a). Moreover, the Emotion by Condition interaction (F (2,96) = 6.59, *p* = 0.0034, ɳ^2^_p_ = 0.120) was also significant and showed a significantly smaller N170 under subliminal condition to sad faces than happy and neutral faces (*p* < 0.01; Figure 5b).

The analysis of N170 peak latencies revealed a significant Emotion × Condition (F (2,96) = 3.69, *p* = 0.029, ɳ^2^_p_ = 0.071) interaction. This effect indicated a shorter N170 latency for both happy and sad expressions (but not neutral) in the supraliminal condition than in the subliminal one (Happy: M = 176.5, SD = 13.6 vs. M = 184.4, SD = 15.9, *p* < 0.001; Sad: M = 180.3, SD = 15.42 vs. M = 186.8, SD = 19.8 *p* = 0.0131; Neutral: 177.7, SD = 17.8 vs. M = 178.3, SD = 14.18, *p* = 0.765; for each emotion comparisons were for supraliminal vs. subliminal condition). Moreover, the interaction of AQ × Facial Gender was significant (F (1,48) = 8.31, *p* = 0.0086, ɳ^2^_p_ = 0.147). This effect showed that in Hi-AQ participants there were no differences between female and male faces (M = 181.1, SD = 13.7 vs. M = 182.3, SD = 12.9, *p* > 0.05), while in Lo-AQ participants, female faces had longer latency than male faces (M = 180.6, SD = 14.8 vs. M = 176.1, SD = 10.5, *p* < 0.05).

On the whole, the present N170 amplitude and latency findings were not consistent with our third and fourth hypotheses.

#### 3.2.3. N2 Amplitude and Latency

The analysis of the midline N2 amplitudes yielded significant interactions of Facial-Gender × Emotion (F (2,96) = 8.00, *p* = 0.0025, ɳ^2^_p_ = 0.143) and of Facial-Gender × Emotion × Location (F (6,288) = 6.08, *p* < 0.001, ɳ^2^_p_ = 0.114). These effects indicated that for happy female-faces there was a larger frontocentral N2 than for male-faces, while for sad faces there was an opposite trend between female and male faces (all *p* < 0.05; Figure 6).

The ANCOVA for N2 latency found a significant main effect of Location (F (3,144) = 62.52, *p* < 0.001, ɳ^2^_p_ = 0.57) and Condition (F (1,48) = 35.76, *p* < 0.001, ɳ^2^_p_ = 0.427). The Location effect showed a progressive significant reduction in N2 latency from Fz and Cz to Pz and Oz regions (all *p* < 0.001). The Condition effect indicated that subliminal stimuli elicited shorter N2 latencies than supraliminal stimuli (M = 225.6, SD = 10.8 vs. M = 209.8, SD = 10.1, *p* < 0.01). No other main or interaction effects were significant.

The analysis of the midline N2 amplitudes yielded significant interactions of Facial-Gender × Emotion (F (2,96) = 8.00, *p* = 0.0025, ɳ^2^_p_ = 0.143) and of Facial-Gender × Emotion × Location (F (6,288) = 6.08, *p* < 0.001, ɳ^2^_p_ = 0.114). These effects indicated that frontal-central N2 to happy female-faces was larger than happy male-faces, while N2 to sad female-faces was smaller than sad male-faces (all *p* < 0.05; Figure 6). 

For N2 latency, we found a significant main effect of Location (F (3,144) = 62.52, *p* < 0.001, ɳ^2^_p_ = 0.57) and Condition (F (1,48) = 35.76, *p* < 0.001, ɳ^2^_p_ = 0.427). The Location effect showed a progressive significant reduction in N2 latency from Fz and Cz to Pz and Oz regions (all *p* < 0.001). The Condition effect indicated that subliminal stimuli elicited shorter N2 latencies than supraliminal stimuli (see Table 2). No other main or interaction effects were significant. The above reported results are new and not in line with our fifth hypothesis.

#### 3.2.4. P3 Amplitude and Latency

Statistical analysis on P3 amplitude scores yielded a significant Location effect (F (3,144) = 9.45, *p* = 0.0019, ɳ^2^_p_ = 0.164), showing larger P3 waves in the Pz and Oz regions than Fz and Cz regions (Fz: M = −0.7, SD = 1.5; Cz: M = 1.3, SD = 1.8; Pz: 3.0, SD = 1.8; Oz: M = 5.4, SD = 3.2; all *p* < 0.001). Further, the following interlinked interactions were all significant: Emotion × Location (F (6,288) = 4.59, *p* = 0.0022, ɳ^2^_p_ = 0.087), Emotion × Location × AQ (F (6,288) = 3.22, *p* = 0.0147, ɳ^2^_p_ = 0.062) and Facial-Gender × Emotion × Location × AQ (F (6,288) = 3.58, *p* = 0.0032, ɳ^2^_p_ = 0.069).

The first interaction showed a larger occipital P3 to sad faces than neutral and happy faces (M = 5.0, SD = 3.3 vs. M = 5.2, SD = 3.4, *p* > 0.05; M = 5.2, SD = 3.4 vs. M = 6.5, SD = 3.3, *p* < 0.05; M = 5.0, SD = 3.3 vs. M = 6.5, SD = 3.3, *p* < 0.01; respectively for happy vs. neutral, neutral vs. sad, and happy vs. sad faces). The second interaction disclosed that Hi-AQ had a larger P3 at occipital lead to sad faces than Lo-AQ (M = 3.4 SD = 1.4 vs. M = 2.2, SD = 1.5, t = 2.92, *p* < 0.01; respectively). The last interlinked effects indicated that for male happy and sad faces, Hi-AQ participants elicited a larger parietal occipital P3 than Lo-AQ ones, while the difference between AQ groups was significant for sad female faces alone (all comparisons survived to FDR *p* < 0.01; see Figure 7).

The P3 latency analysis showed a main effect of Condition (F (1,48) = 20.48, *p* < 0.001, ɳ^2^_p_ = 0.300), indicating significantly shorter P3 latencies in the subliminal than supraliminal condition (M = 288, SD = 12.4 vs. M = 309, SD = 13.8). The Location main effect was significant (F (3,144) = 56.15, *p* < 0.001, ɳ^2^_p_ = 0.539). The Location effect indicated that P3 latencies in Fz and Cz regions were significantly longer than those in Pz and Oz, as well as that in Pz was longer than in Oz (all *p* < 0.001; Table 2). Finally, the significant Condition effect showed that there was a robust P3 latency reduction in subliminal compared to supraliminal condition (Table 2), a result that is opposite to the fifth hypothesis.

#### 3.2.5. N4 Amplitude and Latency

There were no significant main effects for N4 amplitudes, with the exception of Location (F (3,144) = 9.68, *p* < 0.001, ɳ^2^_p_ = 0.168), showing a larger N4 wave in Fz and Oz regions. However, the analysis for the N4 latency found a significant Facial-Gender main effect (F (1,48) = 6.08, *p* = 0.020, ɳ^2^_p_ = 0.112), and a significant interaction Facial-gender × Location interaction (F (3,144) = 9.11, *p* = 0.0007, ɳ^2^_p_ = 0.159), and indicated a significantly shorter N4 wave to female faces than male-faces in Pz and Oz recordings (Pz: M = 384, SD = 16.3 vs. M = 388, SD = 11.5, *p* < 0.05; Oz: M = 350, SD = 28.5 vs. M = 364, SD = 26.8, *p* < 0.05; respectively for female faces vs. male-faces). The Emotion main effect was also significant (F (2,96) = 4.83, *p* = 0.0131, ɳ^2^_p_ = 0.091) and disclosed a longer N4 latency to sad faces than happy and neutral faces (M = 384.3, SD = 9.8 vs. M = 378.2, SD = 11.5 and M = 376.1, SD = 10.2; respectively, both *p* < 0.05).

## 4. Discussion

In the present study, we found no evidence for a significant relation of AQ with RAPM. This result is not new and in line with previous observations indicating no relation between composite AQ and RAPM [83]. We think that this lacking relation can be due to the fact that AQ total score is a composite of facets, such as social skill subscale and attention switching subscale, that are conceptualized as directly and inversely related to RAPM [83]. However, the above-mentioned lacking association makes us exclude general intelligence as a potential factor influencing any significant effect found for AQ. 

Behaviorally, we found that the Hi-AQ group (vs. Lo-AQ) had a reduced accuracy in the detection of facial expressions and that subliminal faces had a lower accuracy relative to supraliminal ones. The Hi-AQ group (but not Lo-AQ) was more accurate to detect facial expressions presented with images of female faces than with male faces, and this facial gender difference was more pronounced for subliminal than supraliminal stimuli (see Figure 2). These findings are aligned with those previously reported in TD individuals showing that a selective impairment in identification of emotional facial expressions is primarily related to the extent of autistic traits [63,69]. 

The current findings are in line with clinical studies on emotional expression processing in people with high-functioning ASD, showing a decline in recognition mainly for negative emotions as disgust and anger [84] and sadness [11]. The authors of these studies suggested that the limited experience in social interactions is a likely source of the observed altered affective behavior in ASD. Although in the present study we cannot exclude this possibility, we had no a priori reason to assume any such differences. Our participants were healthy female psychology students with no history of neurodevelopmental or psychological disorders, which might cause a different way of engaging in social interactions. Further studies might help us validating this assumption. Nevertheless, the lower accuracy in the detection of facial expression between high than low AQ scorers share behavioral similarities with people with autism: Baron-Cohen, Wheelwright [19] also found higher AQ scores among ASD individuals. However, in terms of individual differences in facial gender recognition, the present findings are new and indicate that in women with higher autistic-like traits, female faces facilitate in identifying facial expression.

Consistent with our prediction, the N1 peak amplitude did not change across emotions in Hi-AQ scorers, whereas in Lo-AQ ones, we found that at central and right-parietal regions, the N1 peak to sad faces was significantly higher than that to happy and neutral faces (right quadrant of Figure 3). This early N1 amplitude difference observed in the Lo-AQ participants may have been due to a difference in perception of sad faces rather than due to an early attention effect on face recognition, since it was independent from the presentation time of the face stimuli (i.e., subliminal or supraliminal). To our knowledge, this is one among few studies providing neurophysiological evidence of altered early visual processing of perceived emotional faces in individuals with autistic-like traits. Importantly, this finding is consistent with and extends previous ERP findings by Fujita, Kamio [41] that were obtained in high-functioning ASD individuals, as well as behavioral findings reported in ASD [38,39]. Additionally, we found a longer N1 latency at occipital midline region in Hi-AQ relative to Lo-AQ individuals. This finding was in line with previous Fujita, Yamasaki [40] findings in ASD patients (see Figure 4) and provides novel evidence that not only ASD patients but also TD individuals with autistic-like traits may have weak neural processing of face stimuli. In ASD patients, the inefficient face processing has been speculated to occur because of an impairment in processing chromatic stimuli that preferentially activate the P-color pathway [40]. Therefore, we speculate that this process may be extended to individuals with autism spectrum traits, although perceptual and attentional processing are not independent of each other. Even very early feed forward visual processing cannot bypass top-down control or attentional set, as directly evidenced in ERP studies with a high temporal resolution of brain activities see e.g., [85]. Thus, we maintain that the prolonged N1 latency in Hi-AQ scorers may be part of a broader autism phenotype rather than categorically present for individuals with ASD [19,86]. 

Further, research has demonstrated a magnocellular dysfunction in autism [87,88], and that in terms of cortical processing, the inability to process early visual information correctly should also be taken in high regard in terms of a dysfunctional magnocellular system [89]. The magnocellular pathway, known to be more sensitive to stimuli of lower spatial frequencies [90], activates a subcortical visual pathway that bypasses the visual cortex and has a faster conduction speed than the parvocellular pathway, which is more sensitive to stimuli of higher spatial frequencies [91,92] and dominates input to the dorsal cortical stream. Research has also shown that fast magnocellular projections link early visual and inferotemporal object recognition regions with the orbitofrontal cortex and amygdala and facilitate object recognition by the activation of fast-attentive responses involved in early predictions about objects [89,93].

In terms of N170 wave, our current findings were not in support of our third hypothesis of a smaller and delayed N170 wave in Hi-AQ relative to Lo-AQ scorers [54,62,72] and a larger N170 to sad faces than happy faces [36,70]. We failed to find any effect involving AQ on the N170 amplitude, while this measure was smaller to sad faces than happy and neutral faces under subliminal condition. Yet, we obtained that in Lo-AQ individuals (but not in Hi-AQ ones), emotional recognition of female faces produced a longer N170 latency than male faces. This is a new result that is aligned with previously reported N170 findings in youth and adults with ASD ([49,52,53], e.g., [54,94,95], but see [96] for a contray account) and suggest that this ERP component reflects non-specific configural and attentional processes associated with encoding of structural facial gender cues, rather than with emotional significance per se [97,98]. These novel findings warrant validation. These present findings together with those of accuracy and N1 response ones are also aligned with behavioral and ERP findings for autistic-like traits in general population [69,99] and with clinical studies showing an impairment in ASD patients to recognition of emotional expressions as negative emotions of disgust and anger [84,100] and sadness [11].

The findings of reduced accuracy in the recognition of facial expressions together with longer N1 latency at occipital region and larger P3 amplitude to sad faces in Hi-AQ relative to Lo-AQ scorers indicate that two distinct neural processes may account for dysfunctional facial expression processing in autism-like traits. The first may involve the function of the magnocellular system responsible for early attentional processing, and the latter marks global processing and attention allocation to facial stimuli and is implicated in the integration processing of negative facial expression as sadness. These findings have an important clinical implication since they appear in line with reduced attentional control in autism [87,88,89,97]. These exploratory findings, if being replicated, imply that N1 latency and P3 amplitude parameters might have a possible role as neurophysiological markers of clinical severity of autistic and sensory symptoms. Combining the latency of N1 and P3 to emotional backward-masked faces as stimuli, together with behavioral accuracy and AQ trait score, might have a potential predictive value to assist for clinical diagnosis of autism in adults. However, these novel findings need to be validated in independent samples to test their specificity to ASD diagnosis. 

Finally, in terms of N2 amplitude scores, we failed to support our fifth hypothesis according to which differences between individuals with lower and higher autistic traits would emerge under subliminal viewing conditions (Vukusic, et al., 2017), since we did not find any significant effect involving AQ and/or subliminal/supraliminal condition factors. We found, instead, that frontocentral N2 wave to happy female-faces was larger than happy male-faces, while for sad expression, there was an opposite trend between female and male faces (see Figure 5). Nonetheless, the N2 latency and P3 latency were both shorter for subliminal vs. supraliminal stimuli across both AQ groups. We also found a significantly shorter N4 wave to female faces than male faces in Pz and Oz recordings and to happy faces than sad faces. 

On the whole, Hi-AQ, compared to low AQ scorers, had both higher and longer N1 peaks in the frontal-central leads of the scalp, and a larger parieto-occipital P3 for happy and sad male faces, while differences between AQ groups was significant for sad female faces alone. These findings were seen possibly to reflect more effortful compensatory analytical strategies used by our participants, with high levels of autistic traits to process facial expression and emotion, and support abnormal ERP findings of facial emotion observed within the first 300 ms of stimulus onset in autistic children, which would likely disrupt the development of normal social-cognitive skills [50]. The present results also parallel recent reports by Stavropoulos and colleagues [62] of delayed ERP components in individuals with high AQ scores, and are seen as indicating an inefficient social perception in individuals with subclinical levels of social impairment. Finally, our finding of relative longer N170, N2, and P3 to subliminal vs. supraliminal faces, is consistent with Vukusic and collaborators’ findings [63].

One potential limitation of this study is that we cannot investigate potential effects of gender since the participants in the current sample were female. Since ASD is a predominantly male disorder known to manifest sex differences in face perception (Coffman, et al., 2015), it is worthwhile for future investigations of AQ score on conscious versus nonconscious face processing to analyze male and female data separately. The present sample was drawn from a participant pool of neurotypical right-handed university women students, thus it will also be important to determine that this observed relationship holds in a more diverse population with a more broadly distributed range of traits on the autism spectrum or even an ASD diagnosis.

In sum, behavioral accuracy, N1 latency and N2 and P3 amplitudes were all sensitive to facial gender in the recognition of facial expression. These findings appear in line and complement previous behavioral reports, e.g., [101,102]. Above all, it is important to note that facial gender effects occurred regardless of the task requirement to explicitly attend the gender of the face of each emotional expression and that N1 latency and N2 and P3 amplitude reflect different stages of information processing in facial expressions. First, N1 latency findings indicated that that signals of different facial gender can be discriminated from each other as early as 80 ms following stimulus presentation, a finding that is consistent with previously reported findings on emotional facial expressions [103], showing that signals associated with different facial identities can be discriminated from each other as early as 70 ms following stimulus presentation. Next, N2 amplitude and P3 component are shown to contribute information to both emotional facial expression and facial-gender discriminability. The N4 latency, sensitive to facial gender alone, may reflect more general categorization processes. These effects are compatible with previous ERP results, reporting enhanced activities beyond 200 ms post-stimulus at lateral posterior sites during explicit judgments of facial gender [104,105,106].

## Figures and Tables

**Figure 1 jcm-09-02306-f001:**
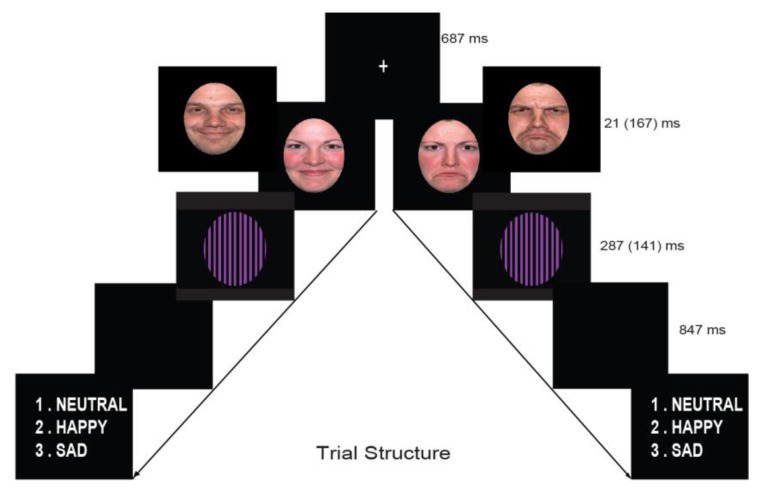
Backward mask paradigm: Stimulus presentation sequences for happy and sad female/male faces (supraliminal time in brackets). A white fixation cross appeared centrally, lasting for 687-ms, followed by a face stimulus (displayed for 21-ms in the subliminal condition or 167-ms in the supraliminal condition). The mask was presented for 287-ms (subliminal) or 141-ms (supraliminal), to keep the presentation time constant for 308-ms.

**Figure 2 jcm-09-02306-f002:**
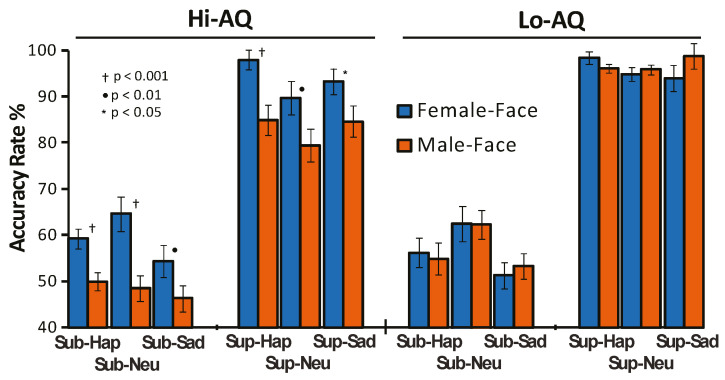
Mean performance values of accuracy scores across subliminal (Sub) and supraliminal (Sup) stimuli of happy (Hap), neutral (Neu), and sad (Sad) female and male faces in Hi-AQ (N = 25) and Lo-AQ (N = 25) women.

**Figure 3 jcm-09-02306-f003:**
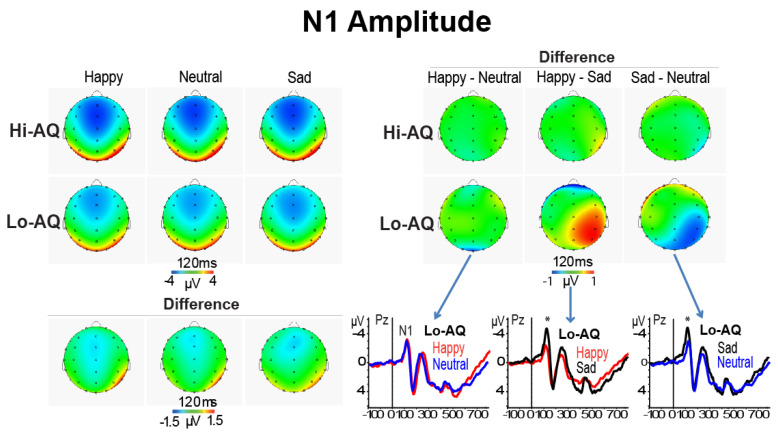
Left-panel: scalp maps and difference maps of N1 amplitude in Hi-AQ (N = 25) and Lo-AQ (N = 25) women. Right-panel: difference maps between emotions separately within Hi-AQ and LO-AQ group. Bottom panel (b): ERP waveforms of emotions in the Lo-AQ group (* *p* < 0.05).

**Figure 4 jcm-09-02306-f004:**
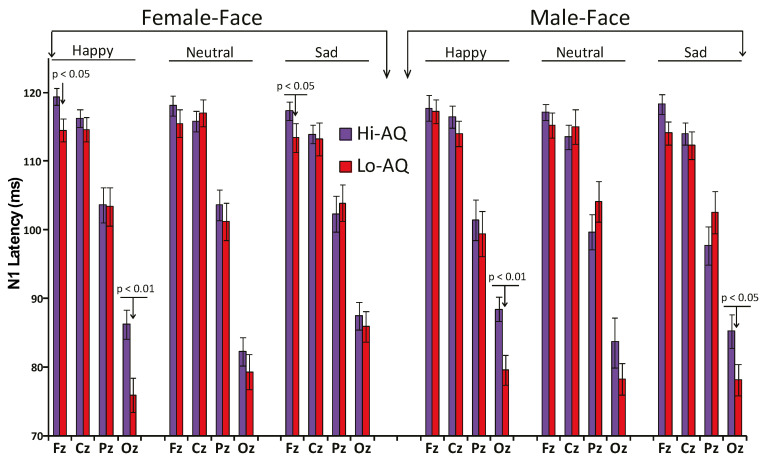
N1 peak latency across midline scalp sites (Fz, Cz, Pz, Oz) to female and male faces of happy, neutral and sad facial expressions in Hi-AQ (N = 25) and Lo-AQ (N = 25) women.

**Figure 5 jcm-09-02306-f005:**
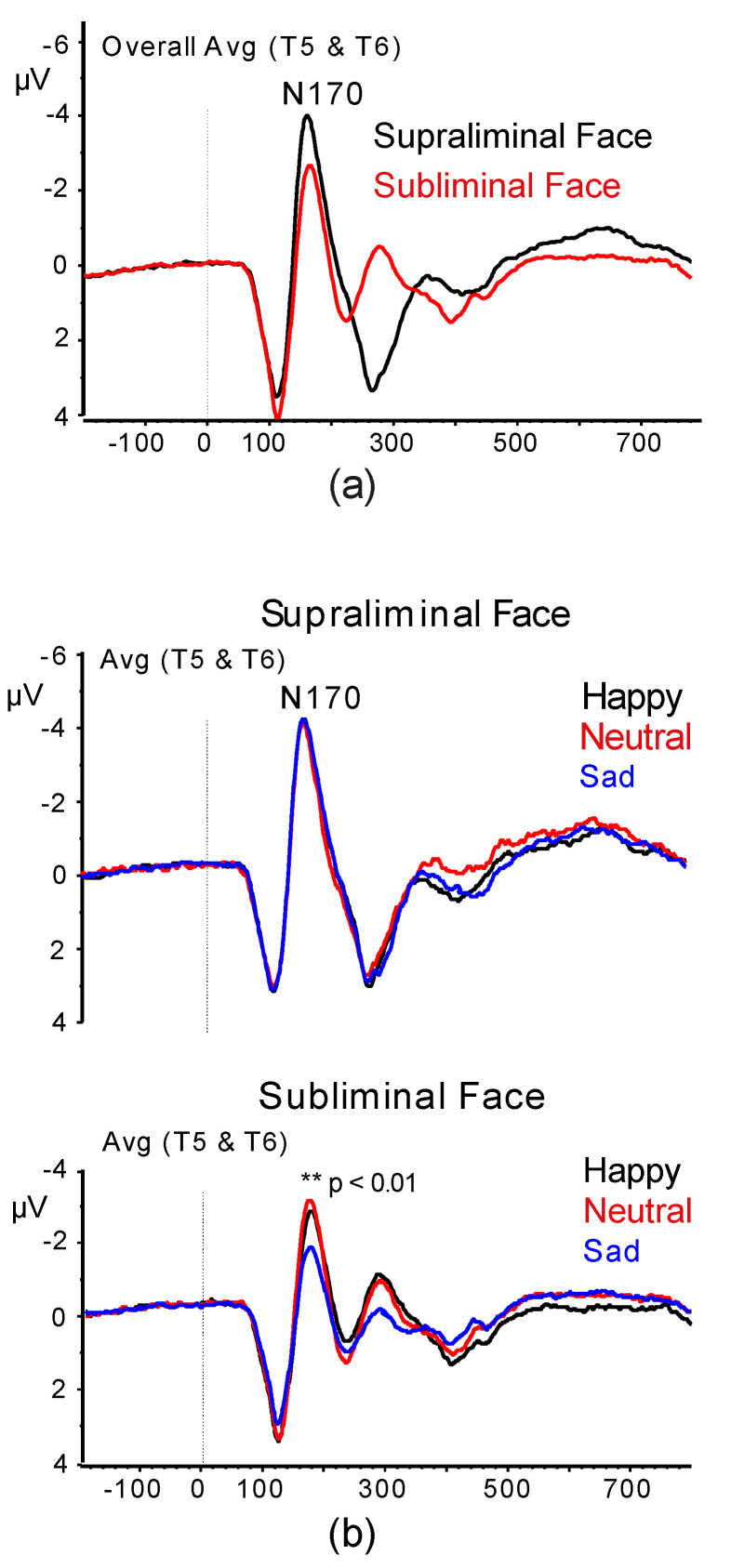
Grand-average ERP waveforms showing the N170 wave at lateral occipital-temporal sites: (**a**) for supraliminal and subliminal conditions, indicating larger waves in the supraliminal than subliminal condition; (**b**) for supraliminal and subliminal conditions of happy, neutral, and sad faces, showing a smaller N170 wave to sad relative to neutral and happy faces.

**Figure 6 jcm-09-02306-f006:**
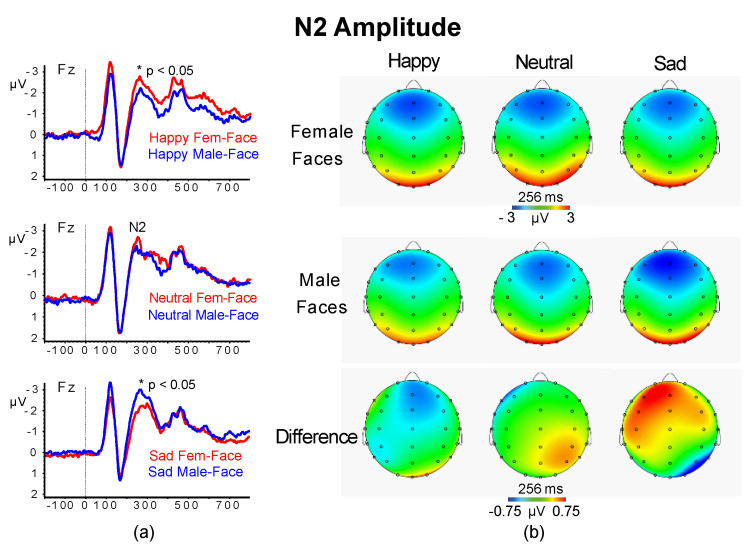
ERP responses at frontal lead Fz (**a**) and scalp maps with difference maps of N2 amplitude for female and male faces of happy, neutral and sad faces (**b**).

**Figure 7 jcm-09-02306-f007:**
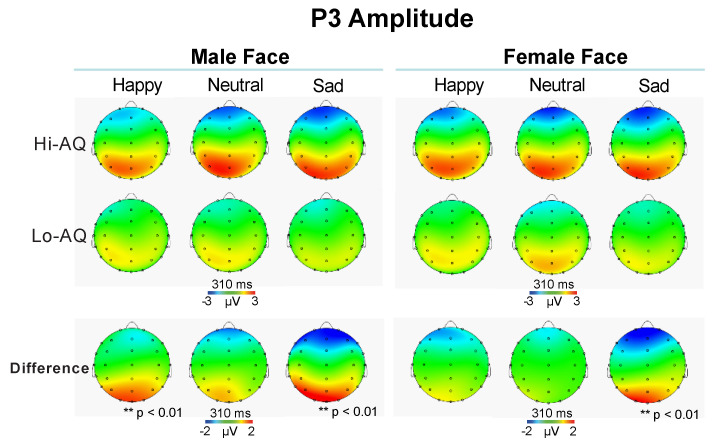
Scalp maps with difference maps of P3 amplitude of Hi-AQ (N = 25) vs. Lo-AQ (N = 25) women for male (**left panel**) and female faces (**right panel**) of happy, neutral and sad emotional expressions.

**Table 1 jcm-09-02306-t001:** Pearson correlations and descriptive statistics for AQ, RAPM and Age scores in 50 women.

	AQ	RAPM	Age
AQ	1		
RAPM	−0.034	1	
Age	0.144	−0.192	1
Mean	14.9	22.5	22.5
SD	7.2	4.7	3.1
Range	3–26	14–34	18–30

Note: Personality Measures - AQ: Autism Spectrum Quotient; RAPM: Raven’s Advanced Progressive Matrices.

**Table 2 jcm-09-02306-t002:** ERP peak latencies (N = 50 women) across all electrodes and separately for each electrode, showing significant differences between subliminal and supraliminal conditions. Probability levels are corrected using False Discovery Rate (FDR) method.

ERP Peak Latencies (ms)	Subliminal	SD	Supraliminal	SD	*p* Values (FDR Correction)
**N170**					
(T5, T6)	183.2	14.5	178.2	13.8	0.0019
T5	188.3	20.3	184.4	20.6	0.066
T6	178.1	12.9	172	12.6	<0.001
**N1**					
(Fz, Cz, Pz, Oz)	102.6	5.7	105.4	5.5	0.0019
Fz	114.5	7.5	118.4	8.4	0.0019
Cz	112.9	8.3	116.4	8.3	0.0019
PZ	102	12.7	102.8	10.7	0.648
Oz	81.1	9.4	84.0	9.8	0.0456
**N2**					
(Fz, Cz, Pz, Oz)	210	10.1	225.6	10.7	<0.001
Fz	235.9	10.9	251.3	13.3	<0.001
Cz	234.1	11.4	251.2	13.8	<0.001
Pz	207.8	23.1	224.1	24.3	<0.001
Oz	161.5	16.2	175.7	16.4	<0.001
**P3**					
(Fz, Cz, Pz, Oz)	288.5	12.4	308.8	13.8	<0.001
Fz	314.6	15.2	327.9	12.9	<0.001
Cz	315	17.8	334.5	19.6	<0.001
Pz	288.3	25.0	306.3	21.9	<0.001
Oz	238.1	19.9	266.7	25.2	<0.001
**N4**					
(Fz, Cz, Pz, Oz)	382	12.2	382.4	10.5	0.822
Fz	393.8	5.8	393.2	4.4	0.453
Cz	392.9	5.6	392.4	6.1	0.515
Pz	383.8	18.7	387.7	13.3	0.186
Oz	357.5	34.4	356.5	31.3	0.866

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
