# Peer review of "Event-Related Potential to Conscious and Nonconscious Emotional Face Perception in Females with Autistic-Like Traits"

_jcm, 2020, doi:10.3390/jcm9072306_

Round 1
Reviewer 1 Report
Thank you for asking me to take a look at the revised version of this manuscript. I think the authors have done a great job of addressing the points I have raised and I have no further suggestions/concerns.
Author Response
We are glad for your appreciation of our revision and that to all the points you raised have been addressed.
Thank You.
Reviewer 2 Report
This is a very interesting study and the authors have done a terrific job in developing and implementing this project. I was also very impressed by the amount of data they managed to gather at the outset with such a considerable sample size. I can appreciate the many hours they must have put into this project not least in data collecting, data entry, and the final analysis process. Although this is an advantage of the present study simultaneously this could be considered as a shortcoming. It was difficult to stay focus on the aim of this manuscript.
In sum, there are two main issues which make this paper a difficult one to read is that there is so much information I doubt that might be possible to fit it into one paper. There were over 8 pages of the presented results which make it very difficult to stay focused on the aim of the study which was simply cited as an exertion of the available data on facial recognition and to understand the impacts of facial gender as a less considered factor among female sample with different levels of AQ!
The second issue is with the format and the way that the data is presented and I think it is because of the amendments that the authors done after the previews comments of the reviewers which changed the referencing system and format. I think a reordering of both parts will be very helpful.
I also found the following points as issues which need to be addressed:
- The style of referencing was very odd to me. The system was different from two aspects first the referral style and second the sequence of the references number from 2 to 8 and then to 14.
- I suggest transferring the presented information on the aim on page 4 from line 133 to 148 to the end of the introduction part. Aims should be clearly cited and the justification for considering the final aims should be presented in the review of the literature part.
- On line 155 page no 4 under the subtitle of Hypothesis:
Hi-AQ should be Lo-AQ
- On line 176 under the subtitle of participants (and also in line 516 of the discussion part)
I do not understand what the adjective "Healthy" referring to! Were they mentally healthy or physically? Was being unhealthy as an exclusion criterion for your sample recruitment? I think "Neurotypical" is a better substitution. Or typically developing as it is mentioned on line 183 (albeit, "typically developing" is a term which is hardly applicable with a sample whose mean age is 23 years old!). Therefore I still suggest using Neurotypical.
- I also do not understand why some very important results presented in Table 2's caption and I recommend bringing most of the presented discretion (from line 3 onward) to the result part's text.
- Although there is a load of presented statistical data and findings in this very lengthy paper, based on my curiosity I was keen on reading any justification regarding the reason for having no correlation between general intelligence score (obtained through RAPM) and other under-investigated factors in the discussion part.
Author Response
Authors’ response to the Reviewer#2
Comments and Suggestions for Authors
This is a very interesting study and the authors have done a terrific job in developing and implementing this project. I was also very impressed by the amount of data they managed to gather at the outset with such a considerable sample size. I can appreciate the many hours they must have put into this project not least in data collecting, data entry, and the final analysis process. Although this is an advantage of the present study simultaneously this could be considered as a shortcoming. It was difficult to stay focus on the aim of this manuscript.
In sum, there are two main issues which make this paper a difficult one to read is that there is so much information I doubt that might be possible to fit it into one paper. There were over 8 pages of the presented results which make it very difficult to stay focused on the aim of the study which was simply cited as an exertion of the available data on facial recognition and to understand the impacts of facial gender as a less considered factor among female sample with different levels of AQ!
--ANSWER: We have reported that the initial aim of the study was to extend the available data on facial recognition by including the impacts of facial gender as a less considered factor among female samples with different levels of AQ. This simple fact has produced several interactional results which force us in a great job reporting all the findings which, in turn, make the paper more difficult to follows. Anyway, we think that they are interesting and merit to be published.
The second issue is with the format and the way that the data is presented and I think it is because of the amendments that the authors done after the previews comments of the reviewers which changed the referencing system and format. I think a reordering of both parts will be very helpful.
I also found the following points as issues which need to be addressed:
- The style of referencing was very odd to me. The system was different from two aspects first the referral style and second the sequence of the references number from 2 to 8 and then to 14.
- -- ANSWER: Referencing was done using the EndNote system.
- I suggest transferring the presented information on the aim on page 4 from line 133 to 148 to the end of the introduction part. Aims should be clearly cited and the justification for considering the final aims should be presented in the review of the literature part.
- On line 155 page no 4 under the subtitle of Hypothesis:
Hi-AQ should be Lo-AQ
- -- ANSWER: Done
- On line 176 under the subtitle of participants (and also in line 516 of the discussion part)
I do not understand what the adjective "Healthy" referring to! Were they mentally healthy or physically? Was being unhealthy as an exclusion criterion for your sample recruitment? I think "Neurotypical" is a better substitution. Or typically developing as it is mentioned on line 183 (albeit, "typically developing" is a term which is hardly applicable with a sample whose mean age is 23 years old!). Therefore I still suggest using Neurotypical.
- -- ANSWER: Thank You. “Healthy” and “typically developing” have been changed with Neurotypical
- I also do not understand why some very important results presented in Table 2's caption and I recommend bringing most of the presented discretion (from line 3 onward) to the result part's text.
- -- ANSWER: Done.
- Although there is a load of presented statistical data and findings in this very lengthy paper, based on my curiosity I was keen on reading any justification regarding the reason for having no correlation between general intelligence score (obtained through RAPM) and other under-investigated factors in the discussion part.
- -- ANSWER: We have now reported in the initial section of the Discussion our justification regarding the reason for having no correlation between RAPM and AQ scores.
We hope that all the points you raised have been addressed with success.
Thank You.
This manuscript is a resubmission of an earlier submission. The following is a list of the peer review reports and author responses from that submission.
Round 1
Reviewer 1 Report
Thank you for asking me to review this manuscript describing a study that examines the relationship between non-conscious processing of facial expressions and ASD traits in a sample of typically developed students using subliminal-supraliminal facial stimuli and ERP. Differences in performance between high and low AQ scores were identified. Additionally, a number of latency differences emerged in relation to subliminal versus supraliminal faces and type of expression, independent of AQ status.
This is an interesting study, particularly as it does not dichotomise ASD but instead considers ASD-related traits in the population at large, and the brain-based mediators between traits and task performance. Whether or not this is relevant to our understanding of the brain basis of ASD is questionable (and ideally should be discussed in the discussion section); consequently, as the potential for knowledge translation is unclear, I am not convinced this is the most suitable journal for this piece of work.
I raise here a number of points that I think can be addressed without too much effort. The language itself is fine, and the style scholarly, but fundamentally this manuscript is very dense and difficult to follow. The introduction is too long, and the results much too detailed. In contrast, the discussion does little to expand on the significance of the results to ASD traits in the population nor ASD itself. I would suggest, therefore, that in any revision of this manuscript, an effort to simplify the text is made, and its clinical relevance is discussed.
For example, towards the end of the introduction you state your hypotheses, but there are so many embedded in just two sentences that it becomes very confusing. Perhaps list your hypotheses and present as ‘primary hypotheses’ and ‘secondary hypotheses’.
When presenting the results, you should extract the primary hypotheses and present those data first and then you can highlight any secondary findings. As it stands, the results are far too detailed and difficult to follow.
Table 1 is helpful but a little confusing . For example, how can you have a mean and SD for gender?
One important finding is that “lower accuracy in Hi-AQ scores compared to Lo-AQ ones”, which you reference as being in figure 2 but this is not so.
An example of confusing text is “and that the N1 and N170 peak latencies were 434 the only latency measures sensitive to the interaction of AQ with consciously/non-consciously 435 presented stimuli and recording region.” The reader is left wondering “…so were latencies longer in higher AQ as you had hypothesised?” Why not just state “consistent with hypothesis one…etc etc”, and refer to figures where appropriate.
As indicated above, the discussion needs some work to highlight the clinical significance of the findings. There is a lot that can be said here in relation to ASD, diagnosis and, perhaps, therapeutics. This does not have to be speculative, just exploratory in light of the findings.
Reviewer 2 Report
In this manuscript, the authors report the results of a study in which participants were presented with subliminal and supraliminal face stimuli depicting emotions (happy, neutral, sad), while their brain activity was measured with electroencephalography (EEG). The measures of interest include accuracy (raw score, d', and c) as well as amplitude and latency to visual Event Related Potentials (ERPs : N1, VPP, N2, and P3). Participants sub-clinical level of autistic traits is also measured with the Autism-spectrum Quotient (AQ). The questions of interest are (1) whether accuracy in determining the emotion of interest differs between subliminal vs. supraliminal and between participants scoring high vs. low on the AQ, and (2) whether visual ERPs to emotional faces are affected by subliminal vs. supraliminal presentation, and differ between participants scoring high vs. low on the AQ. Although those are questions of interest, their novel aspects are not exactly clear to me (the authors mention a very similar study in the introduction : Vukusic et al., 2017) and the analyses performed do not answer them optimally in my opinion, which render the interpretation of the results difficult and suceptible to extrapolations. I have outlined my comments below. I hope that the authors will find them useful and use them to improve the quality of their manuscript.
Introduction :
-
p3, l.95 : How does this research design differ from the one selected by Vukusi et al. (2017) ? Is it simply the facial expressions that differ (i.e. happy, fearful, neutral vs. happy, sad, neutral) ? It would help to be more specific regarding the novel aspects of this research study.
-
p3, l.97 : The sentence starting by « previous » is not clear : do these ERPs reflect processing of subliminal emotions (this is what it seems given that the sentence l101 starts with « by contrast, supraliminal perception... ») ?
-
l.99: Is N2 affected by all emotions vs. neutral or rather by some specific emotions vs. neutral (e.g., fear) ? This may be worth specifying given that the present study does not include fearful face stimuli in its design.
-
l.104 : N4 is introduced here but has not been mentioned earlier. Has it also been linked to conscious perception of emotional faces in past research ? Or looking into this ERP is one of the novel aspect of this research ?
-
l.107 & l.111 : The predictions regarding differences in ERPs between participants scoring high vs. low on the AQ are not well justified theoratically. The expected results would be consistent with previous literature yes, but it would be helpful if the authors could expand on the reason why the results are expected to go in that direction. At present the hypotheses seem to be completely data driven and lack in theoretical motivation.
-
l.111 : The way in which the frontal P3 differs from the typical P3 should be outlined, and the reason for focusing on the frontal P3 explained.
-
l.112 : The design of the present study is not optimal to investigate gender differences (unequal N, see next section), but even if it was the hypotheses regarding the possible impact of gender are extremely vague (how exactly do the authors expect that gender will influence their behavioural and EEG results?), only mentioned in the last sentence of the introduction. Investigating the influence of gender on the effects of interest seem to be an unecessary addition to already complex questions, but if it is to be investigated, the authors should provide some theoretical motivation for it and outline previous literature prompting the addition of this factor.
Method :
-
As previously the sample size of female vs. male is unequal, and the ANCOVA analysis used assumes equal size of the samples of interest, so as homogeneity of variance is required for such analysis. Therefore I strongly advise against including a gender effect in the analysis.
-
Some participants met the screening cutoff for autism (AQ>26 : Woodbury-Smith et al., 2005), as the AQ scores ranged from 3 to 39. How many participants did ? Were those participants referred for further assessment to verify that they did not meet clinical criteria for autism ?
-
Why participants had to take an empathy quotient test (EQ) and an intelligence test (RAPM) ? There are no detailed explanation of what these questionnaires are assessing and no hypotheses pertaining to the influence their scores have on the effects of interest. Thus, it seems that they are unecessary for this study. If those were collected in view of a different analysis, this should be specified.
-
Were luminance and contrast equated across the different emotions ? If not, differences between the emotions could be driven by these low level properties at the ERP level, given that visual ERPs, and particularly early ones (e.g., N1) are sensitive to contrast and luminance (Luck, 2005). Could the authors perhaps acknowledge this as a limitation and/or try to equate low level stimulus properties (using for example the Shine toolbox [Willenbockel et al., 2010]) ?
-
Regarding the procedure, was the presentation time for the fixation cross jittered ? Could the authors expand on literature showing that 21ms is perceived as subliminal and 167ms as supraliminal (justification of timing) ? Is it not problematic that the mask will be presented for different amount of time depending on whether the condition is subliminal or supraliminal (if the purpose of masking is to block emotion processing, then emotion processing will be blocked to a different degree depending on whether the condition is subliminal or supraliminal, introducing an additional difference between those condition)? Is it typical to have such a long delay between the end of the mask and the task ? The black screen following the mask is presented for a very long time (847ms).
-
I am a little puzzled as to why different measurement of accuracy (raw accuracy, d') were used (2.6). Either one is justifiable but one should be chosen as when multple strategies are used, it gives the impression of cherry picking.
-
I do not think that the ANCOVA should include stimuli gender and participants gender as factors (see Intro [7], and Methods [1]). There are no precise hypothesis regarding interactions with these factors, and this unecessarily complicate the interpretation of findings (2.6 & 2.7).
-
l.193 : The reference [81] is repeated multiple times.
-
The ERPs of interest are well described, but were the peaks identified manually or automatically ? Given the number of ERPs looked at, there is a possibility of false positive : was the FDR applied for each ERP separately or across ERPs ?
-
It is a little problematic to use AQ as a continuous measurement for the behaviour but to use groups (with low & high AQ scores) for the ERPs. Ideally, one would like to relate the behaviour to ERP findings. In addition, no information is given relative to these groups (average AQ scores, SD, gender distribution), and to the median score used for the splitting (how many participants were excluded due to having their scores falling on the median ? Were those also excluded for the behavoural analysis?). Given that the participants were recruited from the Psychology department (where students typically have higher social skills: Baron-Cohen et al., 2001), I suspect that the distribution was somewhat skewed toward low AQ scores (is it the case?), which render the splitting by group a little artificial.
-
Why were the Pearson correlations across questionnaire computed ? What is the added information for the question of interest ? What hypothesis is tested with this analysis ?
-
The results are difficult to interpret due to the added effects of participants' gender and stimuli gender but if those are omitted, there could be some interesting behavioural effects. Regardless, the stats are completely missing from the text. This is not acceptable and should be changed. Regarding the figures, it is unclear which differences are significant when the p-value is indicated, the two conditions compard should become visually obvious.
-
For the ERP analysis, I recommend selecting an electrode at which the particular EP of intrest was shown to be maximum in a previous study, or to average the measurement across a few electrodes at which a particular ERP is typically measured, so as to be able to conduct analyses with more interpretable findings. At present, the interactions with gender and electrodes cloud the results pertainig to the question of interest. In addition, as for the behaviour, stats should be provided in the text and effect sizes as well.
-
The figures should somehow illustrate the observed effects. For instance, I would have liked to see for Fig.3 that all emotions are compared in one plot for the high AQ group and in another for the low AQ group.
-
I find the discussion a little unfocused at present but this is mainly because the added factors in the analyses that make interpretation difficult.